# Understanding learning through uncertainty and bias
Rasmus Bruckner [1,2] ✉, Hauke R. Heekeren[1,3] & Matthew R. Nassar [4,5]

Learning allows humans and other animals to make predictions about the environment that facilitate adaptive behavior. Casting learning as predictive inference can shed light on normative cognitive mechanisms that improve predictions under uncertainty. Drawing on normative learning models, we illustrate how learning should be adjusted to different sources of uncertainty, including perceptual uncertainty, risk, and uncertainty due to environmental changes. Such models explain many hallmarks of human learning in terms of specific statistical considerations that come into play when updating predictions under uncertainty. However, humans also display systematic learning biases that deviate from normative models, as studied in computational psychiatry. Some biases can be explained as normative inference conditioned on inaccurate prior assumptions about the environment, while others reflect approximations to Bayesian inference aimed at reducing cognitive demands. These biases offer insights into cognitive mechanisms underlying learning and how they might go awry in psychiatric illness.

Decision-making in an uncertain and changing world depends critically on beliefs about unobservable factors that affect the outcomes of our decisions. Many beliefs, for example, about the quality of a bakery, are learned from past experiences. Since such beliefs are typically based on incomplete and imperfect information, most of our decisions take place under considerable uncertainty[1,2]. For instance, if we would like to get a pastry, we have to deal with uncertainty about the various options, including the quality of potential bakeries or offered pastries at an individual bakery. Uncertainty may arise from limited experience with the different bakeries, variability in the quality of the offered options, or systematic changes, such as when a bakery hires a new pastry chef. To accurately evaluate and choose among the different options, we have to properly consider our limited knowledge about the situation.

Normative Bayesian models offer a prescriptive approach to understanding how humans and other animals should learn and act in uncertain and changing environments for optimal decision-making. A Bayesian modeling approach encompasses a broad spectrum of cognitive processes, from basic behaviors like classical conditioning[3] to complex forms of reasoning, such as social decision-making[4,5] and financial choices[6]. Such models require knowledge of the generative structure of the environment and use Bayes' rule to infer hidden environmental variables based on observed data. When thinking about the quality of different pastries, Bayes' rule prescribes how different experiences should be incorporated into our beliefs about the underlying quality of each option. Such models view previous experiences as probabilistic cues that can be combined to form precise beliefs about the world. This means, for example, that we can estimate the quality of a pastry we have tried many times more reliably than a pastry we have tasted only once. Bayesian models learn probability distributions over possible states of the world that can be used to estimate the probabilities of outcomes contingent on actions. Given such probability distributions, actions can be selected to optimize any cost or objective function, for example, maximizing the expected value of long-run returns, minimizing regret, or simply getting the tastiest pastry.

## Deciding how much to learn

Learning can be cast as a predictive inference problem, where the brain uses a new experience to update beliefs that serve to predict future outcomes[7–9]. One strategy for doing this involves updating predictions according to prediction errors that result from subtracting the prediction from what was actually observed. A primary advantage of error-driven learning is that it is computationally cheap. It only requires storing a single prediction in memory, computing a prediction error, and updating the original prediction by some fraction of the prediction error[10]. For example, eating a pastry tasting much better than predicted would thus elicit a positive prediction error that would increase our assessment of the quality of the bakery.

Within this learning framework, a key question is how much should we update our predictions in response to a prediction error? Formally, the rate at which new prediction errors change previous predictions is called the

[1]Department of Education and Psychology, Freie Universität Berlin, Berlin, Germany. [2]Institute of Psychology, University of Hamburg, Hamburg, Germany. [3]Executive University Board, University of Hamburg, Hamburg, Germany. [4]Robert J. & Nancy D. Carney Institute for Brain Science, Brown University, Providence, RI, USA. [5]Department of Neuroscience, Brown University, Providence, RI, USA. ✉e-mail: rasmus.bruckner@fu-berlin.de

learning rate. A high learning rate means relying almost exclusively on the most recent experience, whereas a lower learning rate puts a greater emphasis on previous experience. Optimal predictions can be approximated in this framework through moment-to-moment adjustments of the learning rate according to different forms of uncertainty[8,11].

In this review, we illustrate that examining human learning through the lens of normative predictive inference sheds considerable light on the dynamics of human behavior. We incorporate studies that directly use Bayesian computational models examining the impact of uncertainty on learning. We also discuss work that relies on reinforcement learning approaches[10,12,13], which can, under specific conditions, approximate principles of Bayesian learning (Box 1). Deviations from normative models can be used to define human learning biases, and the field of computational psychiatry examines such biases in clinical populations. Through this lens, we examine the possibility that many learning impairments in psychiatric disorders might result not from a failure to learn but from a misunderstanding of how much should be learned. Finally, we will argue that the current perspective on learning biases in the computational literature is too narrow. We will make the case that biases can also be the result of simplified learning strategies that approximate optimal inference at a reduced computational cost.

## Decomposing uncertainty

There are multiple forms of uncertainty with dissociable effects on learning, particularly (i) perceptual uncertainty, (ii) risk and estimation uncertainty, and (iii) uncertainty related to environmental changes (Fig. 1). Perceptual uncertainty arises from uncertain perceptual information. For example, imagine visiting a bakery in a foreign country. Being unfamiliar with the local pastries, the displayed options might look very similar to you, making it difficult to distinguish between them (e.g., choosing between a croissant and a Franzbrötchen—a German pastry similar in appearance to a croissant but with a cinnamon-sugar filling; Fig. 1a, b). Risk is defined as the variance of outcomes within a given category, for example, minor variations in the taste or texture of a croissant that we might experience on different occasions that we try it (Fig. 1c). Since risk renders each outcome an unreliable predictor of the underlying quality of the pastry, it leads to estimation uncertainty, that is, an uncertain estimate of the pastry's average or true quality, especially when we have tried it only a few times (Fig. 1d, e). Finally, we will pay attention to the effects of environmental changes, such as shifts in the bakery's ingredients or a new pastry chef, which substantially elevate estimation

uncertainty (Fig. 1f–i). All of these sources of uncertainty affect learning and decision-making, both in theory and practice, and we examine these relationships in detail.

## Perceptual uncertainty prescribes slower learning

Essentially all learning processes rely on sensory information, which is often ambiguous itself or corrupted by internal noise during sensory processing[14–16]. Therefore, perceptual uncertainty is present during learning when the available sensory information is unclear. Returning to our bakery scenario, imagine you are unsure what pastry you have chosen (e.g., croissant vs. Franzbrötchen). When updating the corresponding expected value about how much you like it, perceptual uncertainty leads to a credit-assignment problem between the available options and the outcome of a choice (e.g., the taste of the pastry). In this case, you may confuse the pastries; for example, you eat pastry 1 (croissant) but incorrectly assume pastry 2 (Franzbrötchen) and, therefore, learn the wrong association between the stimulus "pastry 2" and the taste of pastry 1.

To deal with perceptual uncertainty during learning, you can infer a probability distribution over the hidden states underlying the sensory observation, called the belief state[12,13]. For example, based on some ingredients that are familiar to you, you can make an educated guess about the probability of pastry 1 (state 1, croissant) and 2 (state 2, Franzbrötchen). Relying on Bayes' rule, you might infer that pastry 1 (state 1) is 70% likely and pastry 2 (state 2) 30% likely. Belief states that are closer to each other, such as 70% vs. 30%, would reflect that you cannot clearly discriminate the options while more distinct belief states (e.g., 95% and 5%) indicate that they can be more clearly distinguished. Thus, depending on the context, you cannot clearly identify your choice options and experience belief states with different degrees of uncertainty (Fig. 1a, b).

To take into account this sort of credit-assignment problem during learning under perceptual uncertainty, one should consider the belief state. The belief state regulates how much credit each state receives[17]. This can be expressed in terms of the learning rate, which is down-regulated under higher state uncertainty. When you know with certainty that you chose pastry 1 (i.e., belief state signals 100% pastry 1), you can confidently learn the association between stimulus and outcome. However, under high state uncertainty (e.g., belief state signals 60% pastry 1 and 40% pastry 2), you should use a lower learning rate for updating the association between the pastry and taste to avoid learning the wrong association. An emerging line of research examines if humans and other animals take into account

---

## Box 1 | Basics of reinforcement learning models

Reinforcement learning is an overarching framework for solving learning and decision-making problems to maximize reward over time[10]. In psychology and neuroscience, the most popular reinforcement learning model is the Rescorla-Wagner (RW) model, which provides a simple and elegant approach to associative learning. The basic idea is that the model computes the difference between an obtained reward $r_t$ on a given trial and the expected reward or expected value $v_t$, known as the prediction error $\delta_t$:

$$\delta_t = r_t - v_t \qquad (1)$$

and uses the prediction error to update the expected value $v_{t+1}$ for the next trial

$$v_{t+1} = v_t + \alpha \cdot \delta_t \qquad (2)$$

where $\alpha$ denotes the learning rate determining the impact of the prediction error on the expected-value update. The interpretation of

the learning rate can be illustrated based on the rearranged update equation $v_{t+1} = (1 - \alpha) \cdot v_t + \alpha \cdot r_t$, where it becomes clear that $\alpha$ controls the influence of the most recent outcome $r_t$ relative to the prior expected value $v_t$. Greater learning rates ( ~ 1) yield a stronger influence of the most recent reward, while lower values ( ~ 0) lead to a stronger influence of the previous expected value.

The RW model inspired the formulation of several widely used Bayesian models in psychology and neuroscience. For example, the reduced Bayesian model (RBM)[8] and Hierarchical Gaussian Filter (HGF)[51] use update equations similar to eq. (2), where the learning rate is usually a dynamic parameter that changes from trial to trial, in contrast to the RW model with constant $\alpha$. This similarity allows researchers to approximate Bayesian learning with the RW model under certain circumstances. For example, based on the RW model, one can examine learning-rate differences between different environmental conditions, such as frequently changing and stable environments, where a Bayesian perspective predicts higher learning rates in environments with more frequent changes[50,56,57].

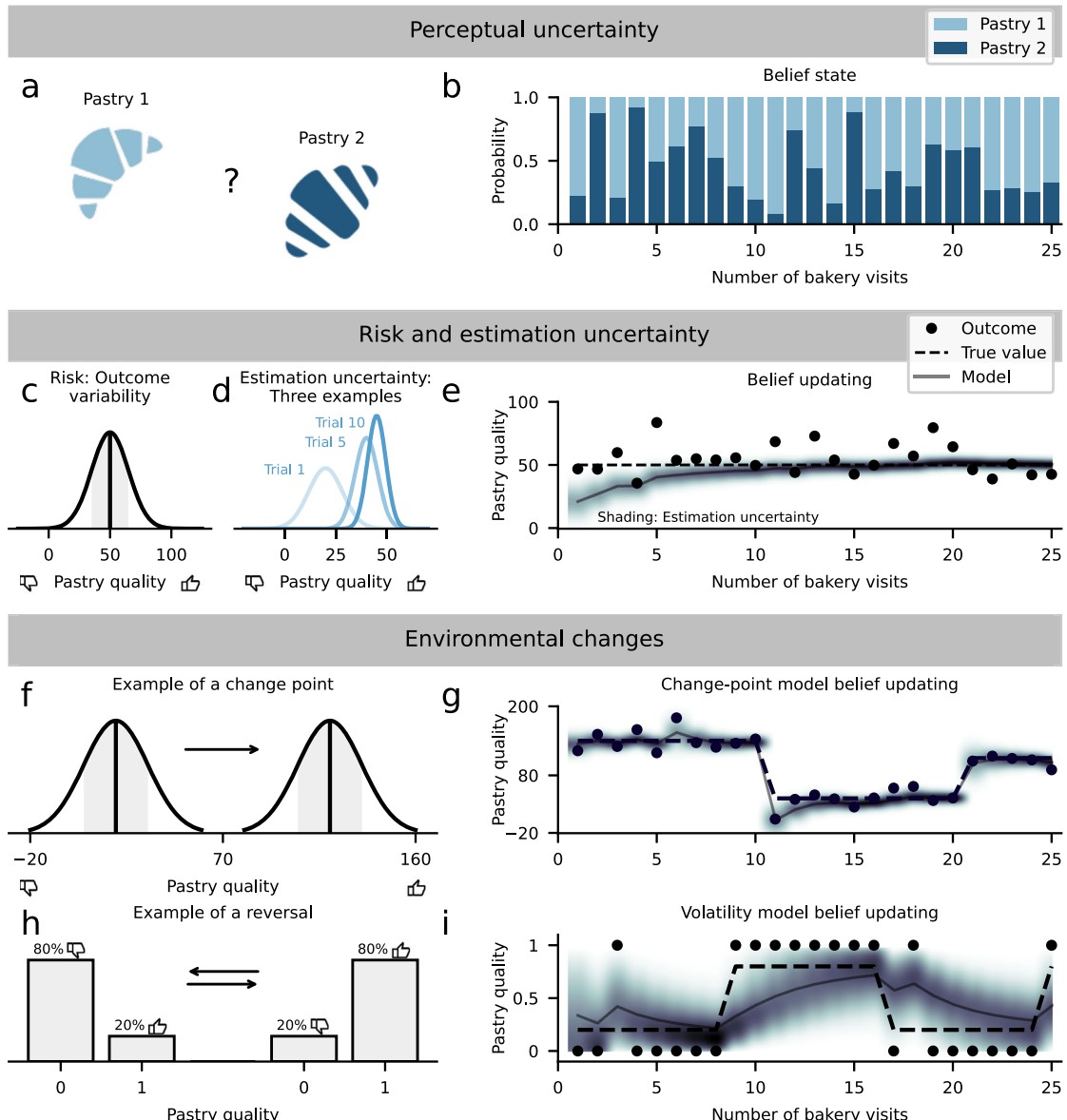

**Fig. 1 | Decomposing uncertainty.** Here, we illustrate three sources of uncertainty: (i) perceptual uncertainty, (ii) risk and estimation uncertainty, (iii) and environmental changes. Perceptual uncertainty. **a** Perceptual uncertainty arises from ambiguous sensory information and internal noise during sensory processing. For example, perceptual uncertainty is present when we cannot precisely distinguish two pastries (pastry 1: croissant, pastry 2: Franzbrötchen—a German pastry similar in appearance to a croissant but with a cinnamon-sugar filling) in a bakery due to similar features like shape and texture. Given the currently available sensory information, evidence of a stimulus or state can be quantified with the belief state reflecting the probability of an uncertain stimulus or state. **b** The plot shows several examples of different belief states about the pastries. The belief state favors pastry 1 on trials 1 and 3, and on trials 2 and 4, pastry 2 is more likely. Trials 2 and 4 are examples of low state uncertainty (>90 % pastry 2), and trial 5 presents an example of high uncertainty (approx. 50 % pastry 1 and 50 % pastry 2). Risk and estimation uncertainty. **c** Risk is defined as outcome variability. For example, if we eat a particular pastry several times, the taste varies to some degree ("Pastry quality"). Thus, although the pastry tastes good on average, natural variability can sometimes lead to worse- or better-than-average outcomes. **d** Estimation uncertainty reflects our imperfect knowledge of the mean underlying value, which emerges from risk because, in the presence of outcome variability, we have to experience multiple outcomes to build up accurate beliefs about the pastry's taste. Therefore, initial estimates of outcome contingencies are more uncertain than estimates based on multiple outcomes. In the pastry example, we have more estimation uncertainty about the average taste after trying it for the first time ("Trial 1") compared to when we have had it multiple times ("Trial 10"). **e** Example of risk (outcome variability) across multiple trials. A normative computational model accurately updates its beliefs about the pastry's quality. Estimation uncertainty is initially high (lighter and broader shading) and decays over time (darker and narrower shading), reflecting that the model's estimate of the taste is becoming more certain. Environmental changes. **f** Environmental changes induce changes in the outcome contingencies of a task. Change points are characterized by sudden changes in the outcome contingencies. For example, the taste of the pastry might systematically improve or worsen when a new pastry chef is hired. **g** Change points facilitate spikes in estimation uncertainty, e.g., because the quality of the pastries produced by the new pastry chef is initially unknown. Change-point models rapidly update their beliefs in response to changes. The lighter and broader shadings after change points illustrate the increase in estimation uncertainty. **h** Another way the environment can change is through reversals between two outcome states. For example, one state in which the pastry tastes bad in 80 % of the time and good in 20 %. After the reversal, the pastry tastes good in 80 % of the cases. In this reversal learning scenario, the outcome contingencies occasionally switch between the two states. **i** In such tasks, volatility models assume gradual changes instead of change points across time. In effect, the model's estimate of getting a good pastry drifts towards the true but unknown outcome contingency of either 80 % or 20 %. Please note that panels (g) (change-point task) and (i) (reversal learning task) show examples of common applications in the literature. A reversal learning task can also be used in combination with continuous outcomes and vice versa. Volatility demonstration based on open-source code by[101], all other simulations based on[102]. Pastry icons were sourced from icons8 (https://icons8.com/).

perceptual uncertainty accordingly, providing evidence that humans indeed adjust their rate of learning about the value of economic choice options, depending on the belief state[17–19].

The above findings suggest belief states guide reward-based learning under perceptual uncertainty, and first studies indicate that one crucial underlying neural mechanism could be the weighting of prediction errors as a function of belief states[20–23]. It is also conceivable that belief-state-based credit assignment involves further inference mechanisms that allow the brain to retrospectively update expected values when perceptual uncertainty is resolved through new information. For example, when perceptual uncertainty about the type of pastry can be eliminated by asking which one it actually is. In such cases, the brain must retrospectively assign credit to the true state. Behavioral work highlighting the ability of people to do such retrospective re-evaluation implies that the brain maintains belief-state and outcome representations over an extended period of time to potentially retroactively re-assign credit when new information is available[24]. The ability to store relevant information for re-assigning credit could dramatically increase the flexibility of learning, and it will be important to fully characterize the limitations of this operation. For example, is it limited to short durations (e.g., seconds, minutes), or can people re-assign credit after longer periods (days, months)?

## Learning under risk and estimation uncertainty

From an economics perspective, we are confronted with risk when outcomes are not perfectly predictable due to outcome variability (Fig. 1c–e)[2,25]. For example, when we get the same pastry several times, it may taste slightly different each time. Mathematically, this can be described by the variance over outcomes, where a larger variance would correspond to higher risk. Because this sort of variability can be learned through observation of multiple outcomes, risk is sometimes also labeled expected uncertainty[26–28]. This economic perspective differs from the definition in clinical settings or what laypeople typically have in mind, where risk often refers to the probability of negative outcomes[29]. Here, we take the economic perspective and define risk in terms of outcome variance.

The next form of uncertainty is estimation uncertainty, which emerges from risk and reflects the learner's uncertainty over the learned or estimated variable, such as the average taste of the pastry[30,31]. If outcomes are variable due to risk, the learner needs to experience multiple outcomes to reliably average out outcome variability. Thus, while risk is a form of uncertainty inherent in the environment, estimation uncertainty is an internal form of uncertainty that comes from the limited samples of the environment that have been observed. As shown in Fig. 1d, estimation uncertainty can be expressed as the variance of the belief distribution over the true but unknown variable that is estimated, such as the pastry's quality in the example. From a normative computational perspective, estimation uncertainty shapes the learning rate by dictating how much a new outcome is used to update the expected value of the choice option. Estimation uncertainty is high when we learn the value of new options and lack relevant outcome experiences to average out outcome variability. Therefore, value estimates are initially unreliable but improve when learning more substantially from new outcome information through a high learning rate. Conversely, after experiencing more outcomes, estimation uncertainty decays, as does the optimal learning rate (Fig. 1 and Box 2).

Converging evidence suggests that humans consider their subjective estimation uncertainty during learning. One line of work has applied tasks with discrete outcomes[31–35]. As described above, in such tasks, risk is reflected in the variance of outcomes, which leads to estimation uncertainty. In these cases, the normative learning rate decays as a function of trials, reflecting the decay in estimation uncertainty. From a statistical perspective, the decay in estimation uncertainty and the learning rate over trials reflects that observers gain a more certain estimate of the outcome probabilities when the estimate is based on more observations. Human participants seem to regulate their learning rates accordingly and tend to use higher learning rates when estimation uncertainty is high.

Other studies have used paradigms with continuous outcomes, where risk is, for example, reflected in Gaussian outcome variability[8,36]. As detailed in Box 2, in such tasks, the combination of risk and estimation uncertainty can be expressed as relative uncertainty, which dictates the normative learning rate. Here the point is that uncertainty about future outcomes is

---

# Box 2 | Equations behind the reduced Bayesian model

The reduced Bayesian model (RBM) provides a way to study how humans adaptively update their beliefs under uncertainty and in changing environments (Fig. 3)[8,47]. The model simplifies a full Bayesian solution to change-point detection[46] for reduced computational complexity and biological plausibility. The update equation is inspired by the Rescorla-Wagner (RW) model:

$$\mu_{t+1} = \mu_t + \alpha_t \cdot \delta_t \tag{3}$$

where $\mu_t$ denotes the model's belief (e.g., about the taste of a pastry), and the prediction error is defined by:

$$\delta_t = x_t - \mu_t \tag{4}$$

expressing the difference between outcome $x_t$ and belief $\mu_t$. The RBM extends beyond the RW model because it relies on a dynamic learning rate that can change from trial to trial depending on uncertainty and environmental changes:

$$\alpha_t = \omega_t + \tau_t - \tau_t \cdot \omega_t \tag{5}$$

The learning rate is composed of two key terms. Change-point probability $\omega_t$ indicates the probability that the last outcome $x_t$ was

generated by a change point ($c_t = 1$) under consideration of the prior probability of a change point (hazard rate $h$):

$$\omega_t = \frac{p(x_t|c_t = 1) \cdot h}{p(x_t|c_t = 0) \cdot (1 - h) + p(x_t|c_t = 1) \cdot h} \tag{6}$$

The parameter is computed based on Bayes' rule, and the learning rate increases with higher change-point probability. This ensures that the model rapidly adjusts its beliefs after a change in the environment. Finally, relative uncertainty $\tau_t$ reflects a combination of risk $\sigma^2$ and estimation uncertainty $\sigma_t^2$:

$$\tau_t = \frac{\sigma_t^2}{\sigma_t^2 + \sigma^2} \tag{7}$$

Relative uncertainty increases in response to a peak in change-point probability and slowly decays as a function of time. The main idea is that the learner's uncertainty over the current belief (estimation uncertainty $\sigma_t^2$) decreases relative to the total uncertainty (sum of estimation uncertainty and risk $\sigma^2$). Therefore, a decrease in estimation uncertainty over trials reflects that the model's belief is getting more and more certain. This leads to a reduction in the learning rate, yielding a weaker influence of new outcomes on the belief update.

---

both due to risk (irreducible outcome variability) and estimation uncertainty (incomplete knowledge about the true outcome probabilities). Therefore, relative uncertainty expresses estimation uncertainty relative to the total uncertainty corresponding to the sum of risk and estimation uncertainty. Higher values indicate that the subjective estimate is quite uncertain and that predictions can be improved through more experience. In contrast, lower values indicate that the subjective estimate is quite certain and that uncertainty about future outcomes is primarily due to risk. In stable or continuously changing environments with these types of outcome contingencies, the learning rate is directly dictated by relative uncertainty, which under such circumstances is known as the Kalman gain[11,37]. This theoretical prediction derived from normative considerations has been supported by several empirical studies showing higher learning rates in less risky conditions[8,30,36,38] and work directly modeling the impact of relative uncertainty on learning, where higher relative uncertainty was associated with higher learning rates[7,30,39].

Finally, when learning about outcome probabilities, choice behavior is not only governed by uncertainty, and there is considerable literature on the effects of positive versus negative outcomes on learning and decision-making[40]. Two prominent effects are the positivity bias (higher learning rate after positive compared to negative outcomes) and the confirmation bias (higher learning rate after a choice was confirmed). While such effects can lead to biases in certain contexts, they might have evolved to strengthen the robustness of learning and decision-making. One line of research argues that positivity and confirmation biases lead to increased differences in reward estimates between options[41–43]. For example, when the true underlying reward probability of a favorable option is 0.6, and the probability of the worse option is 0.4, such biases can increase the subjective difference between them so that, for example, the favorable option has a subjective probability of 0.7. Such an increased expected-value difference can lead to better discriminability between options. This would particularly optimize choice performance when choices are noisy or stochastic. That is, the decision-maker sometimes chooses a sub-optimal option due to random influences, where increased value differences between options would lead to more frequent choices in favor of the best option.

A related line of research shows that asymmetries in responses to positive and negative outcomes might alternatively be computed at the level of the decision-making policy instead of during learning via the learning rate[44,45]. According to the opponent actor learning (OpAL*) model, it might be normative to overemphasize preference differences between different actions[45]. One crucial advantage of this increased discrimination between actions is that choice behavior can be efficiently tuned to the reward environment. Particularly in lean reward environments in which it is challenging to find the best option based on exploration, this policy outperforms strategies without biased policy values. To conclude, both learning-rate and choice biases on the policy level might ultimately serve normative purposes, and as they are not mutually exclusive, might co-exist.

### Environmental changes elevate estimation uncertainty

Changing environments can substantially modulate the degree of estimation uncertainty. For instance, if the bakery hires a new pastry chef, and you have only tried one of their pastries, your estimate of the overall quality is likely to be quite uncertain, but this uncertainty can be reduced by trying more pastries made by the new pastry chef. On top of this, it is often challenging to detect changes themselves, for example, when a pastry is unexpectedly good or bad, but we do not know whether a new pastry chef has been hired. Therefore, in many situations, we are additionally confronted with uncertainty about whether an unexpected outcome is related to a systematic change or merely risk. Normative computational models provide mechanisms to quantify the evidence of a systematic change in the environment and sometimes refer to this as unexpected uncertainty[27,28]. There are two classes of models that are typically used to describe them. One

assumes abrupt changes at environmental change points (change-point models), and the other gradual changes governed by volatility (volatility models). These approaches are not mutually exclusive since many realistic environments can have both abrupt and gradual changes, which can be modeled in a common computational framework[8].

**Change-point models.** Models of the first class presuppose that hidden outcome contingencies jump covertly from one state to another (Fig. 1f, g). An ideal observer model for such environments must consider the possibility that each observed data point reflects a change point (i.e., that each possible pastry was produced by a new pastry chef) and maintain a probability distribution over these discrete possibilities, making optimal inference computationally demanding and intractable for most practical applications[46–48] (Fig. 2). The reduced Bayesian model (RBM) developed by Nassar and colleagues[8] strongly reduces such demands and considers only the possibility that a change point did and did not occur in the most recent trial, thereby indicating the probability that the environment just changed (Figs. 2 and 3). In the RBM, the approximation to optimal inferences allows us to mathematically describe belief updating based on three key terms: prediction error, change-point probability (CPP), and relative uncertainty (RU) (Box 2). In effect, the model predicts that individuals strongly increase their learning rate when change points are more likely (higher CPP) to appropriately adjust to the changed outcome contingencies. Consequently, individuals re-start their learning process and experience a substantial increase in estimation (and relative) uncertainty. In contrast, when change-point probability is low, learning is mainly governed by relative uncertainty as described above (see also Fig. 1; for further related work, see Meyniel and colleagues[34] and Payzan-LeNestour and colleagues[49]).

**Volatility models.** The second class of models assumes that states of interest undergo gradual changes governed by volatility (Fig. 1h, i). Models incorporating such assumptions can capture how humans learn the expected values of the choice options in changing environments[50]. This work provided first evidence that individuals use higher learning rates in a reversal learning task when outcome contingencies change frequently compared to when outcome contingencies are more stable. To efficiently approximate optimal Bayesian inference assuming gradual changes driven by volatility, Mathys and colleagues introduced the Hierarchical Gaussian Filter (HGF)[33,51]. This computational model allows researchers to estimate individual differences in parameters assumed to regulate learning according to volatility. The canonical HGF distinguishes three coupled levels that can be mapped to (i) observed outcomes (e.g., good vs. bad pastry), (ii) estimated outcome contingencies (governing frequency of good vs. bad pastry), and (iii) estimated volatility leading to changes in the outcome contingencies (how the probability of good vs. bad pastry changes). Risk is captured in the first level's outcome probabilities, estimation uncertainty reflects incomplete knowledge about the true but unknown outcome contingencies at the second level, and the third level reflects individual assumptions about volatility leading to environmental changes. Due to the hierarchical nature, the model incorporates two types of learning rates: The outcome-contingency learning rate at the second level regulates belief updating about outcome contingencies, while the volatility learning rate at the third level regulates belief updating about volatility. The HGF has been applied in many computational psychiatry studies[32], as described in more detail in the following section.

### Computational psychiatry

In the previous sections, we have highlighted normative computations that prescribe how people should learn under uncertainty. Computational psychiatry studies (sub-)clinical conditions under which people systematically deviate from normative inference, often by comparing model parameters between groups (Box 3). Currently, the common and often

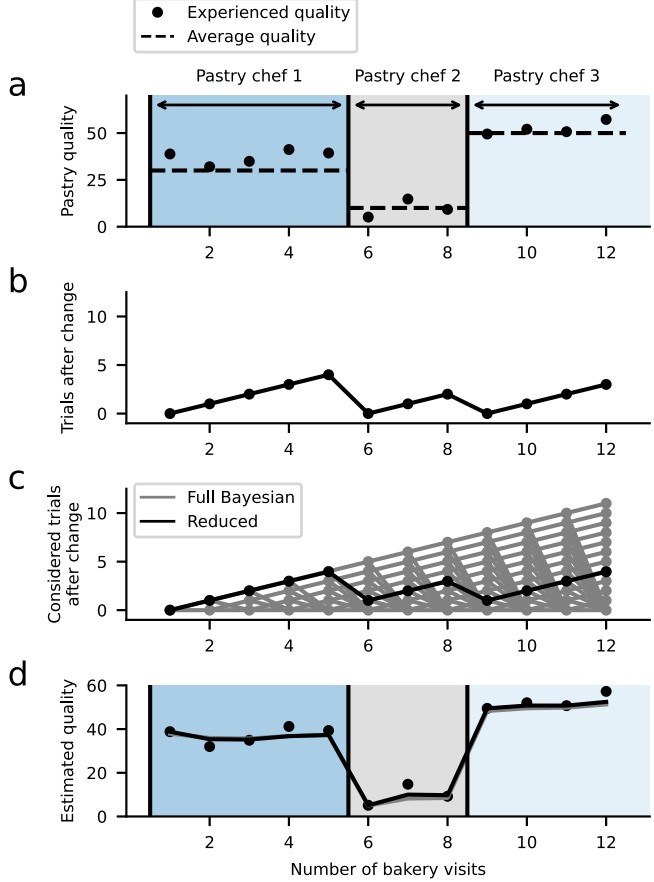

**Fig. 2 | Approximating a normative model of inference amid change points.** Bayesian inference models can often provide insights into human behavior, but they are typically computationally demanding, particularly in discontinuously changing environments, raising questions about how optimal inference might be approximated to achieve normative inference at a reduced computational cost. **a** For example, the food quality of a bakery might randomly vary to some degree (defined as risk) so that each time a customer visits the bakery, the quality might be different to some degree ("Experienced quality"). In addition to risk, the bakery's quality might occasionally change systematically when a new pastry chef is hired, or the menu is updated (change points in "average quality"). **b** If the customer knows when such change points have taken place, the best possible way to estimate the quality would be to compute the average quality since the last change point. **c** However, in the real world, we often do not know if an unexpected event, such as a pastry that is worse than expected, resulted from random variability (inconsistency in ingredients) or a change point (new pastry chef). To estimate the quality in the absence of knowledge of exact change-point locations, an optimal model (gray lines indicating full Bayesian model) must consider the possibilities that a change point did or did not occur on each day (i.e., that each possible pastry was produced by a new pastry chef) and maintain a probability distribution over the growing set of discrete possible run lengths (each reflecting a possible number of days that the current pastry chef has previously baked). To approximate this optimal inference at a small fraction of the computational cost, a reduced Bayesian model (black line) approximates this distribution over run lengths with a single "expected run length" by taking a weighted average of the two possibilities (new pastry chef versus same pastry chef) on each timestep (compare to subplot (b)). **d** Belief updating in the reduced Bayesian model corresponds to an error-driven learning rule with a dynamic learning rate. Full Bayesian model based on[46]; reduced model based on[8,39].

implicit assumption is that these deviations stem from altered priors about the environment, such as an overestimation of the rate of change or volatility. Such studies suggest that dysfunctional learning might often not result from a failure to learn but rather from a misunderstanding of how much to learn. To examine these ideas in more detail, we next give a selective overview of recent work on anxiety, autism spectrum disorder (ASD), and obsessive-compulsive disorder (OCD).

## Anxiety

Anxiety is associated with altered beliefs about threat, particularly higher expectations of aversive outcomes[52]. In some cases, uncertainty might already be sufficient to elicit anxiety[32]. Here, we focus on the effects of anxiety on learning under uncertainty and whether anxiety is associated with altered learning rates. Currently, the literature can broadly be divided into studies suggesting (i) that anxious individuals generally use altered learning rates, (ii) that anxious individuals have difficulties in adjusting learning rates to environmental changes, and (iii) that there is no relationship between anxiety and learning rates.

First, some evidence shows that anxiety is associated with overlearning from aversive outcomes. Participants with unmedicated mood and anxiety disorders showed particularly high learning rates in response to negative feedback[53]. Similarly, there is some evidence for higher lose-shift rates and higher learning rates in anxious individuals[54]. Such findings could suggest that anxiety is associated with higher estimation uncertainty and an underestimation of risk in the environment. This could increase learning rates and lead to stronger beliefs about negative events. In line with this interpretation, self-reported cognitive state- and trait-anxiety symptoms, including worry, are associated with greater estimation uncertainty and subjectively higher threat probabilities[55]. However, different facets of anxiety could also drive different, sometimes even opposing, dysfunctional learning tendencies: Somatic anxiety symptoms, including arousal, were related to lower estimation uncertainty and lower threat probabilities. Similarly, when state anxiety was directly induced through experimental manipulations, participants showed depleted levels of estimation uncertainty, leading to lower learning rates and larger error rates[35]. That is, state anxiety led participants to be more resistant to belief updating in response to new information and nudged them to rely more strongly on prior beliefs.

Second, other work suggests that anxiety is linked to difficulties adjusting to environmental changes, potentially due to a misunderstanding about the degree of volatility. Browning and colleagues[56] argued that subjects with high trait anxiety have more difficulties than healthy controls in adaptively adjusting learning rates according to the level of environmental volatility. As described above, learning rates should be lower in stable environments, while under volatility, higher learning rates are required to adjust to the changes. These learning dynamics were apparent in low-anxious individuals, but subjects with higher trait anxiety showed more comparable learning rates between the environments. There is evidence suggesting that these difficulties are particularly driven by reduced learning rates in response to positive prediction errors under high volatility (where learning rates are ideally high)[57]. Moreover, this work highlights that impairments in learning under volatility arise from internalizing symptoms common to anxiety and depression. Thus, learning biases reported in previous work might not be specific to anxiety, and future research should take similar trans-diagnostic dimensions into account.

Third, more recent studies failed to find a relationship between anxiety and learning. One study induced state anxiety and did not find a relationship between anxiety and choice accuracy and failed to find model-based evidence of an effect of anxiety on learning[58]. Similarly, two studies focusing on trait anxiety and transdiagnostic factors did not find systematic evidence of an association between learning rates and anxious-depression symptoms[59,60]. One potential explanation for these heterogeneous results is that studies have employed different tasks, and results might not generalize across paradigms. Moreover, the applied modeling frameworks often differ considerably. Finally, in many cases, the psychometric properties of tasks and modeling parameters are unclear. For future work, we recommend systematically testing the three hypotheses outlined above across a broad range of task conditions using a unified modeling framework to better understand how anxiety biases learning and whether its effects generalize across task settings.

## Autism spectrum disorder

ASD is a neurodevelopmental disorder with a diverse range of symptoms, including difficulties in social interaction and communication, restricted

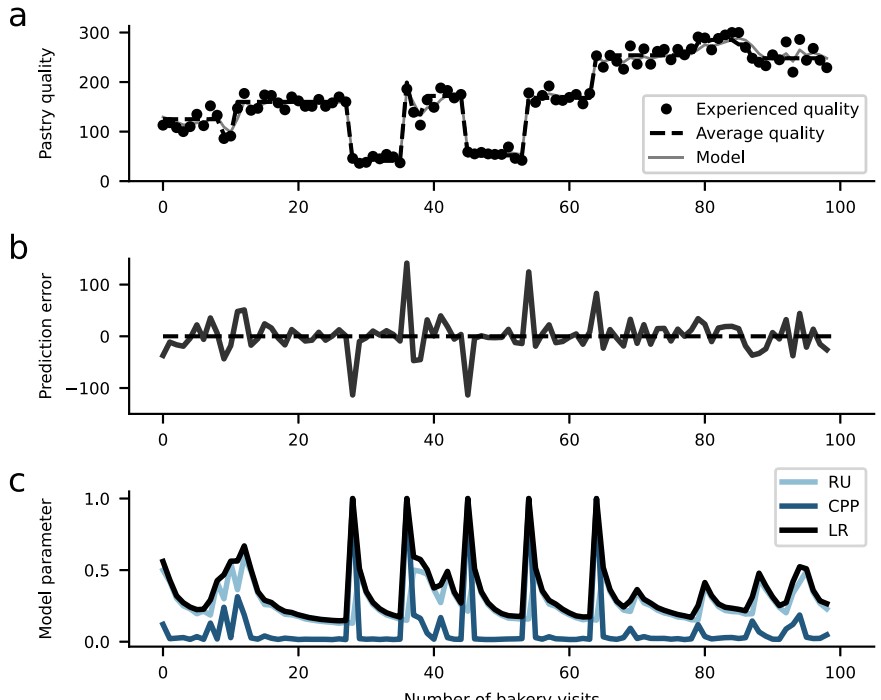

**Fig. 3 | Reduced Bayesian model.** Illustration of learning dynamics of the reduced Bayesian model (RBM) capturing learning in risky and changing environments[8]. **a** As in the example from Fig. 1, outcomes $x_t$ reflect the taste of a pastry ordered several times in a bakery. Due to risk, outcomes are variable from trial to trial and cannot be predicted perfectly. The RBM accurately learns the true but unknown quality over time and rapidly adjusts to change points. **b** The model computes prediction errors reflecting the difference between actual and predicted outcomes. **c** To optimally take risk and environmental changes into account, the model dynamically regulates its learning rate (LR) $\alpha_t$, governing the degree to which it utilizes prediction errors for learning. The learning rate is determined by relative uncertainty (RU) and change-point probability (CPP). RU reflects the model's estimation uncertainty (e.g., about the quality of the pastry) relative to its total uncertainty (sum of estimation uncertainty and risk). When RU is high, it indicates a high contribution of estimation uncertainty to the model's total uncertainty, which yields an increase in the learning rate. When RU is low, estimation uncertainty has a low contribution to the model's uncertainty, and the learning rate tends to be lower, reflecting that the major source of uncertainty is irreducible risk. CPP reflects the probability of a change point on the last trial and leads to an increasing learning rate.

and repetitive behavior, and increased attention to detail[61]. A prominent hypothesis on the computational underpinnings of ASD symptoms holds that deficits in perception, learning, and social cognition result from an imbalance of the precision ascribed to sensory evidence relative to prior beliefs[62–64]. Accordingly, individuals with ASD would expect highly precise sensory input so that newly arriving information overrules prior beliefs during belief updating, akin to a higher learning rate. Currently, the evidence for this hypothesis is mixed, and we here provide a selective overview of modeling studies examining learning and decision-making in ASD (see also systematic and comprehensive review of studies across different cognitive domains by Chrysaitis and Seriès[65]).

Some studies are overall in line with the hypothesis of an imbalance in the impact of prior beliefs and new information on belief updating in ASD. One study shows that individuals reporting increased attention to detail show overlearning in environments with risk and change points[66]. From a normative perspective, this finding suggests that autistic individuals underestimate the magnitude of risk, thereby assuming that outcomes are less heavily distorted by random variability and treating prediction errors as more reliable teaching signals. This result aligns with the intuition that autistic individuals integrate fewer outcomes during learning, reflecting a strong focus on local stimulus details rather than the global average over the history of observations.

Moreover, in reversal learning, autistic individuals might overlearn about the frequency of environmental changes[67]. Based on the HGF, it has been suggested that their beliefs about environmental volatility fluctuate too strongly in response to new outcomes, thereby preventing accurate representations of how much the environment changes. A more recent study with a similar reversal learning task is partly consistent with these results and found higher overall learning rates in children and younger adults with ASD

compared to control subjects. These results were accompanied by higher perseveration rates in ASD participants[68].

Other studies do not support the hypothesis of an imbalance between prior beliefs and newly arriving sensory information. Two studies with reversal learning tasks failed to find evidence of learning-rate differences between ASD and control groups[69,70]. Another study did not find significant group differences in volatility estimates between an ASD and control group but hinted at potential subgroup differences in ASD, which would need to be confirmed independently[71]. Moreover, a study in the general population did not find systematic evidence of a correlation between autistic traits and learning rates[72]. Finally, a developmental study comparing children with ASD and a matched control group suggests similar learning-rate adjustments to different degrees of volatility[73].

Taken together, evidence of systematic learning-rate differences between groups indicating an imbalance between the influences of prior beliefs and new information on belief updating is mixed. We note that some of the studies that do not support the imbalance hypothesis found performance differences between groups, but such differences seem to originate independently of learning rates[70,72]. Crucially, the studies were obtained with different tasks, participant populations, and modeling frameworks. This highlights the need for future work to rigorously examine the psychometric properties of the measures used and to systematically compare tasks to ensure the robustness and generalizability of the results.

**Obsessive-compulsive disorder**

OCD is characterized by an increased performance of habitual, repetitive actions (compulsions), often in response to obsessions, such as intrusive thoughts[74–76]. A computational psychiatry approach can help to identify what mechanisms might give rise to such overly strong habitual actions.

## Box 3 | Estimating free parameters of a computational model

Computational models can be interpreted as quantitative hypotheses about cognitive processes underlying task behavior. We can develop different models, where each represents a distinct hypothesis, and compare such models to each other in light of the observed data. To do so, researchers often rely on parameter estimation or "model fitting" to find the set of parameters that best describes the participants' behavioral data, such as the learning rate or the consideration of uncertainty for learning.

We typically distinguish between a learning model like the Rescorla-Wagner model or the reduced Bayesian model (RBM) formalizing the learning process itself and the observation model linking the learning model's predictions and the observed empirical data. The purpose of the observation model is to assign probabilities to the measured data conditional on the predictions of the learning model. To find the best-fitting model parameters, we can rely on a number of available methods. The most common method is maximum-likelihood estimation. The combination of the learning and observation model defines a likelihood function, and maximum-likelihood estimation allows us to compute the most probable parameter values, that is, the values that maximize the likelihood function. For example, when estimating free parameters of the RBM, the likelihood function might be defined by

$$p^{\varepsilon}(a_{1:T}|x_{1:T}) = N(a_{1:T}; \hat{a}_{1:T}, \varepsilon^2) \qquad (8)$$

where $a$ denotes the empirical data (in this case, the belief updates of the participant) in response to the presented outcomes $x$, and $\hat{a}$ denotes the model's belief updates. $\varepsilon$ is a motor-noise parameter that accounts for the residuals arising from deviations between participant and model behavior, and $T$ refers to the total number of trials. In this example, the RBM corresponds to the learning model based on which we obtain the model's belief updates $\hat{a}$, and the Gaussian in eq. (8) is the observation model linking the RBM to the empirical data (here $a$). The motor-noise parameter $\varepsilon$ accounting for residuals due to imprecise motor control has to be estimated (panel (a) shows three values of $\varepsilon$ for an example trial $t$). Panel (b) shows the log-likelihood of a range of parameter values (blue curve), and maximum-likelihood estimation allows us to find the best parameter value with maximum likelihood (black dot). Beyond the parameters of the observation model, we could use the method of maximum likelihood to estimate parameters of the RBM (i.e., the learning model), such as the influence of change-point probability on learning or the underestimation of uncertainty[30,39].

Finally, comprehensive modeling requires additional analyses and methods, including the systematic comparison of alternative models and different model-validation analyses like parameter and model recovery. Moreover, alternative approaches to parameter estimation are available, such as hierarchical Bayesian parameter estimation. Excellent guidelines and tutorials are provided by references[103,104].

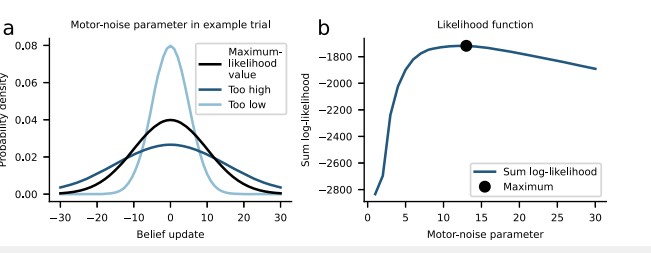

Particularly whether patients overestimate the uncertainty of their beliefs, in line with the intuition that OCD is strongly associated with doubt and intolerance of uncertainty[77]. Currently, research might suggest that patients maladaptively deal with uncertainty, but the replicability of the results and the exact computational origins are debated.

Some evidence suggests that OCD patients use elevated and inflexible learning rates that are not responsive to uncertainty[78]. As such, they fail to dynamically assess how much to learn and resort to high learning rates through which prediction errors strongly drive belief updating, potentially due to an underestimation of risk, subjectively high hazard rates, or higher estimation uncertainty. Moreover, it might fit the idea that patients with OCD have strong doubts about their beliefs and therefore seek to correct them based on the most recent information (e.g., when observing a new outcome), potentially similar to frequent checking behavior common to OCD. However, when the OCD group of this study was asked about their belief confidence, their confidence reports were similar to those of the control group, suggesting that—at least directly after reporting their beliefs —OCD patients do not experience excessive uncertainty over their beliefs that might drive the elevated updates. Yet there is currently also a debate about whether these results are directly linked to compulsivity or, alternatively, co-morbid symptoms also common to generalized anxiety disorder and depression[79] and the degree to which these results can be replicated[80,81]. Together, this line of evidence might suggest that reports of

beliefs and the experienced uncertainty about them are misaligned in compulsive individuals. However, more research is required to better understand if such symptoms are robustly linked to biased learning.

In summary, computational psychiatry studies provide evidence that different patient populations show altered behavior in tasks that require dynamic adjustments of learning in response to uncertainty and environmental changes. However, studies considerably differ with respect to their behavioral tasks, modeling approaches, and study populations, leading to heterogeneous conclusions about the computational underpinnings of altered learning in clinical conditions. This raises the question of whether deviations between clinical and healthy control groups might appear more consistent when studied based on alternative views other than the one discussed so far that assumes altered priors about the environment. Therefore, in the next section, we consider the possibility that approximations to Bayesian learning may explain the origins of learning biases more comprehensively.

## The origin of learning biases

Current computational psychiatry studies often assume that psychiatric symptoms alter a person's priors or assumptions about the environment, thereby changing the regulation of the learning rate. For example, researchers often interpret altered learning in anxiety patients as stemming from patients' assumptions about a quickly changing environment. That is,

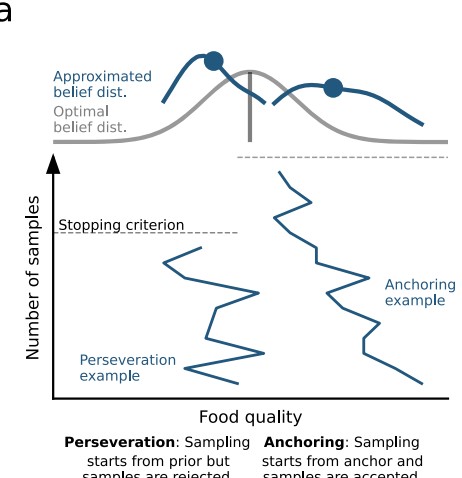

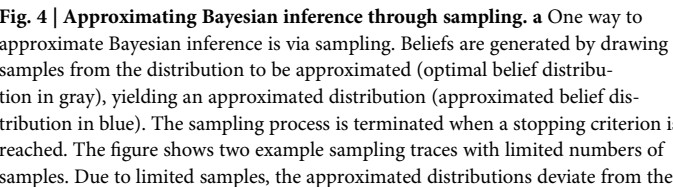

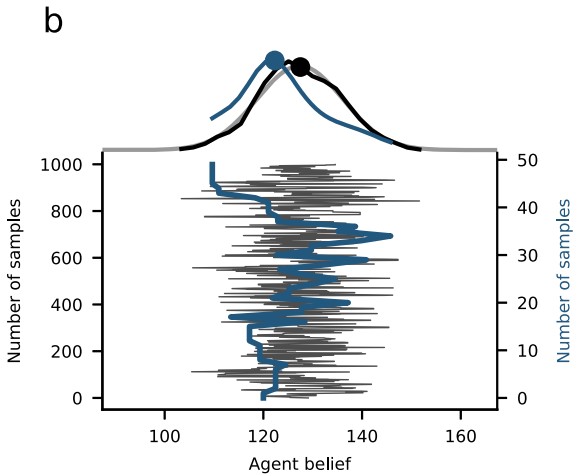

**Fig. 4 | Approximating Bayesian inference through sampling. a** One way to approximate Bayesian inference is via sampling. Beliefs are generated by drawing samples from the distribution to be approximated (optimal belief distribution in gray), yielding an approximated distribution (approximated belief distribution in blue). The sampling process is terminated when a stopping criterion is reached. The figure shows two example sampling traces with limited numbers of samples. Due to limited samples, the approximated distributions deviate from the optional distribution, which can be one source of learning biases. **b** The amount of sampling navigates a trade-off between belief accuracy and computational cost. When many samples are drawn (in this case 1000), the approximation is accurate (compare black and gray curves) but computationally more costly compared to fewer samples. Learning in humans can be described with far fewer samples (e.g., 50 samples), where the costs of sampling are lower and the approximation is less accurate, leading to biased belief updating (compare blue and gray curves).

from this perspective, learning impairments originate from miscalibrated assumptions about the true generative environment. We argue that another potential origin of learning biases lies in the specific learning mechanisms people use to approximate normative computations. Here, we focus on one such approximation to normative learning through "sampling", which is a cognitively plausible mechanism to simplify normative computations in a way that they are more tractable for humans and other animals.

## How approximating normative learning can lead to biases

Learning biases can emerge from strategies that people employ to approximate normative computations. In complex and dynamically changing environments, it is infeasible to rely on exact Bayesian inference because it would require unrealistically high computational resources or might be analytically intractable altogether. Bayes rule can be expressed in the following way

$$p(\mu|x) = \frac{p(x|\mu)p(\mu)}{\int p(x|\mu)p(\mu)d\mu} \qquad (9)$$

where $\mu$ is the variable that we want to infer, such as the quality of a bakery, and $x$ refers to the experience, such as a pastry that we have tried. When conducting Bayesian inference, we are interested in the posterior distribution $p(\mu|x)$. Computing the posterior based on analytic update equations can be challenging when the denominator requires solving integrals that are infeasible or when computations are costly based on numerical approximation methods. Therefore, if the nervous system has evolved to perform normative computations, it has to rely on smart approximations at lower computational costs that do not require explicitly computing the denominator with intractable integrals. Two prominent and psychologically plausible approximation schemes are sampling and variational inference[82,83]. Sampling approximations do not require computing posterior distributions exactly, and instead approximate them by drawing samples. Variational methods replace exact posterior distributions with simpler distributions that are easier to use. These approximation schemes are not mutually exclusive, but here we focus on sampling approximations because they have more consistently been used as a framework for understanding cognitive biases[83].

The basic idea of sampling is that the brain samples from the posterior belief distribution in different locations to approximate it[39,84–88]. Like in Markov Chain Monte-Carlo (MCMC) procedures, the sampling process begins at an initial value and drifts through different hypothetical beliefs, storing plausible values in a chain that extends until a stopping criterion is reached. For example, when thinking about the quality of a new bakery, one might start with an initial hypothesis about the quality of the bakery and begin considering other hypotheses (and storing the plausible ones) after tasting its food (Fig. 4a). By setting a conservative stopping criterion and thus drawing many samples, one can closely approximate the optimal belief distribution (Fig. 4b).

Each sample can be interpreted as a computation that draws on a limited cognitive resource, providing an integrative framework for thinking about the trade-off between performance and compute cost. An advantage of relying on fewer samples is that it might be cognitively cheaper compared to more samples. But on the other hand, drawing fewer samples comes at the cost of biases and less accurate beliefs. Indeed, people are thought to rely on a liberal stopping criterion, thereby drawing fewer samples than what would be required to closely approximate the belief distribution. This strategy would be computationally efficient but lead to systematic biases and higher response variability than predicted by fully normative models. That is to say, by considering the cognitive costs of learning, sampling provides a way to understand learning in terms of resource rationality[89–91].

Through this view, individuals and groups may differ in how they navigate the trade-off between computationally efficient versus accurate inference, potentially providing a framework for better generalizing behavioral differences across tasks or conditions. We have recently proposed a sampling model capturing dynamic learning under uncertainty[39]. The model updates beliefs through a sampling process and relies on a stopping criterion that controls the termination of sampling and commitment to a belief update. One crucial advance is that the model predictions generalize across very different task conditions. When the model relies on fewer samples mimicking fewer cognitive resources, and the stopping criterion is more quickly reached, it predicts biased learning, but the exact type of bias depends on the task condition (Fig. 4a). When beliefs are updated from those formed on a previous trial, perseveration occurs when new samples do not sufficiently increase predictive accuracy, leading to the repetition of the previous prediction. In contrast, when the start point of the updating process is manipulated externally, providing an "anchor" from which the sampling process begins, updated beliefs are biased toward that anchor. The

anchoring bias emerges in particular when the model only draws a limited amount of samples and concludes the sampling process before the new samples can fully compensate for the influence of the external anchor. In typical applications of MCMC, biases due to the start point of the sampling process are usually eliminated by discarding the initial samples, known as the "burn-in period". However, sampling models with anchoring bias, including our own model, explicitly assume that burn-in samples are not discarded, yielding the anchoring bias[39,86]. Thus, a key prediction of the model is that both perseverative and anchoring biases come from the same source, namely the reliance on reduced sampling. Data from human subjects who performed both task conditions support this prediction, showing that the degree of perseveration in one condition predicts the degree of anchoring in the other[39].

In contrast to this, current normative models often lack the capacity to generalize across conditions. While the RBM can successfully explain both perseveration and anchoring in individual task conditions, it can only do so by adding specific parameters designed to capture descriptive features of the observed data. Since these descriptive features differ across conditions, the model fails to generalize predictions from one condition to the other. This is in line with a larger literature highlighting the lack of correlation in parameter estimates obtained by fitting models to the same participants performing different tasks[92]. Thus, the success of the sampling framework described above suggests that resource-rational models could be a useful tool for addressing some of the heterogeneity that has been observed across tasks and modeling frameworks in previous computational psychiatry studies.

Considering different developmental groups as a test case for this idea, children and older adults display hallmarks of a frugal sampling policy that promotes biases in order to limit computational cost[93,94]. We found that the two biases discussed above (perseveration and anchoring bias) are greater in children and older adults compared to their younger counterparts[39]. This pattern could be reproduced in the sampling model with the assumption of less sampling, consistent with the idea that children and older adults have fewer available resources for learning, and adopt a sampling strategy that minimizes cognitive resource expenditure in order to compensate for this.

We are of the opinion that future work should take a sampling perspective towards computational psychiatry. A sampling approach bears the potential to trace several biases back to the cognitive mechanism people rely on for learning. For example, difficulties in responding to changes, which is sometimes assumed to underlie anxiety, can be explained by "under-sampling", where an insufficient amount of samples is drawn to re-calibrate the belief to the new situation. Previous work suggests that such models can generalize across different learning pathologies observed in different task conditions, positioning it well to explain the diverse findings in learning studies within computational psychiatry. Finally, the fact that a sampling perspective directly considers the computational costs of learning would allow us to directly incorporate differences in cognitive resources into computational perspectives on clinical disorders.

## Concluding remarks

Navigating vast environments requires sophisticated learning abilities to constantly update our beliefs about what the future holds. Normative Bayesian models prescribe how we should update our beliefs by optimally taking into account uncertainty. In this review, we have focused on error-driven learning models, where belief updating is driven by prediction errors, signaling how well our predictions match observed outcomes. An essential factor that determines how accurately beliefs are updated is the learning rate, expressing how strongly prediction errors are weighted to update predictions. We have shown that the learning rate critically depends on several coupled yet dissociable forms of uncertainty.

We argued that learning biases, that is, deviations from normative learning, can have at least two origins. First, a misunderstanding of the true generative environment. For example, people might have assumptions about the rate of change or the amount of risk in an environment that do not

match actual environmental statistics. Second, biases can emerge from computational mechanisms that people employ to approximate normative learning. Humans rely on limited cognitive resources and often inappropriate information for optimal reasoning and might, therefore, resort to heuristic strategies that work well in most situations[95]. Similarly, the resource-rationality framework proposes that human reasoning is optimal given limited cognitive abilities and incomplete information[89,90,96]. Here, we have focused on learning through sampling, which has been proposed as a resource-rational strategy for approximating complex Bayesian learning[86,88]. Of note, approximate Bayesian inference does not necessarily guarantee that a complex model is rendered tractable for real-world cognition but provides a promising framework for integrating the role of mental resources into cognitive models[97].

Computational biases might help better understand altered learning and decision-making in anxiety, ASD, and OCD. These studies suggest that different symptoms are linked to altered behavior in tasks that involve adapting learning to uncertainty and changing environmental conditions. However, recent work also shows that the reliability of computational parameters is limited, and the employed tasks often differ substantially. It will be important to systematically take such concerns into account, ideally combined with larger sample sizes[81,98–100].

Finally, a promising approach for future research is linking learning biases in psychopathology to transdiagnostic symptoms. Several of the reviewed studies suggest that computational biases are not necessarily specific to a particular diagnosis, such as OCD and anxiety disorder, but arise across diagnostic boundaries[55,57,79]. An efficient way to conduct such studies with a large number of participants is to rely on online data collection, including transdiagnostic screening instruments.

To conclude, uncertainty is pervasive in learning and decision-making. Combining properly designed experiments with normative modeling approaches can provide insights into how uncertainty should guide learning and how biases in such computations impair adaptive behavior. We propose that a more general view of these biases might emerge through more careful consideration of resource-rational models that consider, quantify, and parameterize the trade-off between the expenditure of mental resources and the achievement of desired task behaviors.

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

## Acknowledgements
We thank Amrita Lamba, Linda Yu, Muhammad Hashim Satti, Prashanti Ganesh, Sienna Bruinsma, and Yuan-Wei Yao for helpful comments on earlier drafts of the manuscript. We also thank Owen Parsons for helpful discussions.

## Author contributions
Rasmus Bruckner played a lead role in writing-original draft and writing-review and editing. Hauke R. Heekeren played a lead role in supervision and a supporting role in writing-review and editing. Matthew R. Nassar played a lead role in supervision and writing-review and editing.

## Funding

## Competing interests
The authors declare no competing interests.
