## [Transparent Peer Review file · Communications Psychology]

Understanding learning through uncertainty and bias

Corresponding Author: Dr Rasmus Bruckner

Version 0:

Decision Letter:

**** Please ensure you delete the link to your author homepage in this e-mail if you wish to forward it to your co-authors ****

Dear Dr Bruckner,

Your Review Article titled "Understanding Learning Through Uncertainty and Bias" has now been seen by 2 referees, whose comments appear below. In the light of their advice I am delighted to say that we are happy, in principle, to publish a suitably revised version in Communications Psychology.

We will handle your revision in-house and not send the revised paper for further review if, in the editors' judgement, the referees' comments on the present version have been fully addressed. Please note that forming this judgment editorially relies on a clear and well-documented response to all concerns. In addition to addressing the reviewers' comments, I ask you to respond to the editorial comments on the edited version of your manuscript.

EDITORIAL REQUESTS:

* Please review the changes in the attached copy of your manuscript, which has been edited for style, and address the comments and queries I have added. If using Word, please use the 'track changes' feature to make the process of accepting your manuscript more efficient.

* Please check whether your manuscript contains third-party images, such as figures from the literature, stock photos, clip art or commercial satellite and map data. If any of the display items in your manuscript (figures, tables, boxes or movies) include images that are the same as, or are adaptations of, previously published images, please fill in the [Third Party Rights Table](http://www.nature.com/licenceforms/snl/thirdpartyrights-table.doc), and return to us when you submit your revised manuscript. This information will enable us to obtain the necessary rights to re-use such material. If we are unable to obtain the necessary rights to use or adapt any of the material that you wish to use, we will contact you to discuss alternative options.

* Communications Psychology uses a transparent peer review system. On author request, confidential information and data can be removed from the published reviewer reports and rebuttal letters prior to publication. If you are concerned about the release of confidential data, please let us know specifically what information you would like to have removed. Please note that we cannot incorporate redactions for any other reasons.

*If you have not done so already, please alert me to any related manuscripts from your group that are under consideration or in press at other journals, or are being written up for submission to other journals (see www.nature.com/authors/editorial_policies/duplicate.html for details).

FORMATTING GUIDELINES:

** Abstract

The paper's Abstract (up to 150 words; without references) should serve both as a general introduction to the topic, and as a brief, non-technical summary of your main points and their implications. It should start by outlining the background to your article (why the topic is important) and the main question you have addressed, before going on to describe your key points, main conclusions and their general implications. Because we hope that researchers across all fields of psychology will be interested in your work, the preface should be as accessible as possible, explaining essential but specialised terms concisely. We suggest you show your Abstract to colleagues in other fields to uncover any problematic concepts.

** Figures

Please remove all figures from the main text and upload them individually, one figure per file. To ensure the swift processing of your paper please provide the highest quality, vector format, versions of your images (.ai, .eps, .psd) where available. Text and labelling should be in a separate layer to enable editing during the production process. If vector files are not available then please supply the figures in whichever format they were compiled in and not saved as flat .jpeg or .TIFF files. If your artwork contains any photographic images, please ensure these are at least 300 dpi.

* Figures should be simple and informative — multi-part figures are best avoided. Boxes should occupy no more than half a page in the PDF (less than 500 words) and may include a figure.

* References

References appear as superscript Arabic numerals, in order of mention. The reference list mentions references in the numerical order in which they are mentioned in the main text. If a reference is cited more than once, the same number is used throughout the text and the reference receives a single entry in the reference list.

We ask that you select the most significant 5–10% of references in your list for highlighting, and add a single sentence in bold after each of these references to describe the main result and its significance.

Only papers that have been published or accepted by a named publication should be in the reference list (preprints and citations of datasets are also permitted). Unpublished/Submitted research should not be included in the reference list; it should only be mentioned briefly and parenthetically in the main text. Note that no major arguments should rely on unpublished research.

Published conference abstracts and URLs for web sites should be cited parenthetically in the text, not in the reference list.

Footnotes are not used.

* Competing interests

Please include a "Competing interests" statement after the References. Note that we ask authors to declare both financial and non-financial competing interests. For more details, see <https://www.nature.com/authors/policies/competing.html>. If you have no financial or non-financial competing interests, please state so: "The authors declare no competing interests."

SUBMISSION INFORMATION:

In order to accept your paper, we require the following:

* A cover letter describing your response to our editorial requests.

* A separate document detailing your point-by-point response to any issues raised by our referees (please include the referees' comments in this document).

* The final version of your text as a Word or TeX/LaTeX file, with any tables prepared using the Table menu in Word or the table environment in TeX/LaTeX and using the 'track changes' feature in Word.

* Production-quality versions of all figures, supplied as separate files. Photographic images should be 300 dpi in RGB format (.jpg, TIFF or native Photoshop format) and any labels/scale bars included in a separate layer from the image. Line art, graphs and schemes should be vector format (.ai, .eps, .pdf); Adobe Illustrator files are preferred and will minimize production time. Any chemical structures or schemes contained within figures should additionally be supplied as separate Chemdraw (.cdx) files.

Communications Psychology is a fully open access journal. Articles are made freely accessible on publication. For further information about article processing charges, open access funding, and advice and support from Nature Research, please visit <https://www.nature.com/commpsychol/open-access>

Please note that your paper cannot be sent for typesetting to our production team until we have received this information; **therefore, please ensure that you have this ready when submitting the final version of your manuscript.**

ORCID

Communications Psychology is committed to improving transparency in authorship. As part of our efforts in this direction, we are now requesting that all authors identified as 'corresponding author' create and link their Open Researcher and

Contributor Identifier (ORCID) with their account on the Manuscript Tracking System (MTS) prior to acceptance. ORCID helps the scientific community achieve unambiguous attribution of all scholarly contributions. For more information please visit <http://www.springernature.com/orcid>

For all corresponding authors listed on the manuscript, please follow the instructions in the link below to link your ORCID to your account on our MTS before submitting the final version of the manuscript. If you do not yet have an ORCID you will be able to create one in minutes.

IMPORTANT: All authors identified as 'corresponding author' on the manuscript must follow these instructions. Non-corresponding authors do not have to link their ORCIDs but are encouraged to do so. Please note that it will not be possible to add/modify ORCIDs at proof. Thus, if they wish to have their ORCID added to the paper they must also follow the above procedure prior to acceptance.

To support ORCID's aims, we only allow a single ORCID identifier to be attached to one account. If you have any issues attaching an ORCID identifier to your MTS account, please contact the [Platform Support Helpdesk](http://platformsupport.nature.com/).

Link Redacted

We hope to hear from you within two weeks; please let us know if the process may take longer.

Best regards,

Marika

Marika Schiffer, PhD
Chief Editor
Communications Psychology

REVIEWERS' EXPERTISE:

Reviewer #1 computational modelling, decision making
Reviewer #2 computational modelling, decision making

REVIEWERS' COMMENTS:

Reviewer #1 (Remarks to the Author):

In this review/opinion paper, Bruckner and colleagues look at learning as a predictive inference problem, and explain different how different types of uncertainty can affect learning in both healthy individuals and also in people with certain mental health diagnosis. The paper is well written and could be useful for scientists studying learning and/or doing computational psychiatry.

I have only a few comments/suggestions:

- I find the example of the two rice bowls not very intuitive. How often is it that we order a bowl of rice and we don't know which specific bowl of rice did we chose? Instead, for example, the person could go to their favorite restaurant that has regularly 2 chefs that rotate, and so when you are ordering the bowl of rice you don't know which of the chefs is working that day (this would be different from the sudden change in chef talked about later in the paper).
- Do the authors have an explanation, using this paradigm, of conditioned taste aversion? If I go to a restaurant for the first time and order a bowl of rice, and then I get sick after, I will most likely never even attempt to sample that particular bowl of rice/restaurant combination again. It is ok if the authors don't (this way of looking at learning doesn't have to explain everything), I was just curious.
- Another thing that the authors could perhaps comment on is the potential effects of rewards and costs on the learning rates.
- The computational psychiatry section is missing some relevant literature. For example, in the "autism spectrum disorder", the way the paragraph is written it seems that only a few studies have been done on the matter. Instead, there is a huge literature on the topic (the authors may be aware of the literature, is just that a reader not familiar with it may get the wrong impression). For a very good review/meta-analysis on the topic, see Chrysaitis and Series, 2023.

- Small suggested change: In the abstract, instead of “Learning allows humans and animals” change to “Learning allows humans and other animals”.

Reviewer #2 (Remarks to the Author):

This review article discusses how humans and animals learn under different forms of uncertainty, such as perceptual uncertainty, risk, and environmental change, using normative Bayesian models as a framework. The authors explore how learning biases arise from either misaligned priors or computational approximations, and provide insights into psychiatric disorders.

I believe this review is highly valuable and useful for those interested in computational perspectives on perception, learning and behaviour. It provides a well-organised summary of topics ranging from traditional frameworks to recent advances, making it a useful resource. Before publication, I would like to ask for a few clarifications, which I have outlined below.

Comment #1:

This review primarily focuses on normative Bayesian models, which infer environmental states through Bayesian inference based on an environmental model. However, particularly in the discussions on the relationship between learning rates and psychiatric disorders, many cited studies appear to use parameter estimation from simpler reinforcement learning models (e.g., the Rescorla-Wagner model), which often assume the absence of perceptual uncertainty. It would be helpful to briefly mention these models and clearly inform readers that not all referenced studies were conducted using Bayesian models. Additionally, it would be beneficial to include an explanation on how behavioral data is fitted to parameters, as this may not be familiar to all readers.

Comment #2:

Regarding The Origin of Learning Bias (p.15-):

I understand that when the number of samples in MCMC is finite, estimation biases can occur. However, if the sampling process is properly conducted, these biases would not consistently skew in one direction but instead fluctuate randomly. This raises the question of how specific directional biases observed in psychiatric disorders can be explained by this mechanism. Perhaps I am overlooking something, but if this issue has not yet been addressed, I would appreciate a discussion on this point.

Minor points:

p.7, Line 29-

“Human participants seem to regulate their learning rates accordingly and tend to use higher learning rates when estimation uncertainty is high...”

I expected that lower learning rates would be used in such cases, rather than higher ones. If my understanding is incorrect, I would appreciate a brief clarification to make this point clearer.

p.11 Box 1

It might be helpful to include a brief and intuitive explanation of why equations (1) and (2) take their specific forms. This would aid readers in understanding the underlying rationale behind these formulations.

p.15 Line 26

“the denominator requires solving integrals that are infeasible or costly based on numerical methods.”

- I am unclear about why calculating this denominator is necessary. If the goal is to determine the relative plausibility of different values of μ , it seems the denominator may not be required, as normalization into probabilities isn't necessary (e.g., as in MAP estimation). It would be helpful if this point could be clarified.

Figure 1a (p.4)

This might stem from cultural differences and may not need much attention, but as a Japanese reader, I found the depiction of the rice bowls a bit strange. The image seems to resemble plain bowls of rice without toppings, which is not typically something one would choose between at a restaurant. Additionally, in Japanese culture, standing chopsticks upright in a bowl is a gesture reserved for offerings to the deceased and is considered highly inauspicious and inappropriate in other contexts.

That said, the authors' narrative is very clear, and the fundamental storyline is well-constructed. I believe it can remain as it is without any major changes.

Review Response

Rasmus Bruckner, Hauke R. Heekeren, Matthew R. Nassar

We thank the reviewers for their positive evaluations. Below, we have included point-by-point responses to each of the reviewers' concerns. Page numbers refer to the revised manuscript. Excerpts from the reviews are in black, replies are in blue, and excerpts from the revised manuscript are *italicized*.

Reviewer 1

In this review/opinion paper, Bruckner and colleagues look at learning as a predictive inference problem, and explain different how different types of uncertainty can affect learning in both healthy individuals and also in people with certain mental health diagnosis. The paper is well written and could be useful for scientists studying learning and/or doing computational psychiatry.

I have only a few comments/suggestions:

- I find the example of the two rice bowls not very intuitive. How often is it that we order a bowl of rice and we don't know which specific bowl of rice did we chose? Instead, for example, the person could go to their favorite restaurant that has regularly 2 chefs that rotate, and so when you are ordering the bowl of rice you don't know which of the chefs is working that day (this would be different from the sudden change in chef talked about later in the paper).

We agree that our original example could have been more clear. We have now replaced it with an example drawn from our own real-life experiences. When co-author Matt Nassar stayed in Germany for a few months in 2024, he often visited a local bakery with a wide array of pastries. He found it challenging to distinguish between a croissant and a Franzbrötchen—a German pastry that is visually similar to a croissant but differs in its cinnamon-sugar filling. This situation arose because the pastries were displayed together, often without labels, and their appearances were quite similar to someone unfamiliar with the local cuisine. This real-world example illustrates how uncertainty can emerge in decision-making, even in seemingly simple choices, and aligns closely with the conceptual framework we discuss in the paper. We now use this example on page 3 (lines 22-35).

For example, imagine visiting a bakery in a foreign country. Being unfamiliar with the local pastries, the displayed options might look very similar to you, making it difficult to distinguish between them (e.g., choosing between a croissant and a Franzbrötchen—a German pastry similar in appearance to a croissant but with a cinnamon-sugar filling; Fig. 1a and b). Risk is defined as the variance of outcomes within a given category, for example, minor variations in the taste

or texture of a croissant that we might experience on different occasions that we try it (Fig. 1c). Since risk renders each outcome an unreliable predictor of the underlying quality of the pastry, it leads to estimation uncertainty, that is, an uncertain estimate of the pastry's average or true quality, especially when we have tried it only a few times (Fig. 1d and e). Finally, we will pay attention to the effects of environmental changes, such as shifts in the bakery's ingredients or a new pastry chef, which substantially elevate estimation uncertainty (Fig. 1f-i). All of these sources of uncertainty affect learning and decision-making, both in theory and in practice, and we examine these relationships in detail.

- Do the authors have an explanation, using this paradigm, of conditioned taste aversion? If I go to a restaurant for the first time and order a bowl of rice, and then I get sick after, I will most likely never even attempt to sample that particular bowl of rice/restaurant combination again. It is ok if the authors don't (this way of looking at learning doesn't have to explain everything), I was just curious.

Thanks for this question. Indeed, we can explain conditioned taste aversion using a Bayesian perspective, like in our manuscript. The basic idea behind Bayesian models of classical conditioning ("latent variable models")¹ is that humans and other animals learn to predict both the conditioned stimulus (CS) and unconditioned stimulus (US) based on the history of experiences. Both CS and US are generated by an underlying latent cause. For example, the CS might be the observable fact of eating a chicken filet, and the unconditioned stimulus might be feeling sick. The latent cause underlying this association could be whether the chicken was contaminated (latent cause $x_t = 1$) or safe (latent cause $x_t = 0$).

The person would aim to predict the current CS_t (chicken filet) and US_t (feeling sick) based on the history of events ($CS_{1:t-1}, US_{1:t-1}$) according to $p(CS_t, US_t | CS_{1:t-1}, US_{1:t-1})$. During inference, this perspective explicitly considers that the observed variables are generated by latent cause x_t :

$$p(CS_t, US_t | CS_{1:t-1}, US_{1:t-1}) = \sum_{x_t} p(CS_t, US_t | x_t) p(x_t | CS_{1:t-1}, US_{1:t-1}) \quad (1)$$

According to this inference model, the person would expect to get sick after eating the chicken as long as latent cause $x_t = 1$ (chicken contaminated) is active. When a different latent cause is clearly active (e.g., at a different restaurant), the person would no longer expect to get sick, as their inference takes into account the new context, which suggests a different underlying cause for the observed food.

Since this Bayesian perspective on classical conditioning has been covered elsewhere in detail (e.g., Courville et al. (2006)¹), we decided to keep our discussion on this brief and added the following (page 2, lines 14-16):

A Bayesian modeling approach encompasses a broad spectrum of cognitive processes, from basic behaviors like classical conditioning¹ to complex forms of reasoning, such as social decision-making^{2,3} and financial choices⁴.

- Another thing that the authors could perhaps comment on is the potential effects of rewards and costs on the learning rates.

Thanks for the suggestion. Rewards and costs have relevant effects on the learning rate, and two lines of research have interpreted these as normative influences. We have added two paragraphs discussing these findings in detail (page 9):

Finally, when learning about outcome probabilities, choice behavior is not only governed by uncertainty, and there is considerable literature on the effects of positive versus negative outcomes on learning and decision-making⁵. Two prominent effects are the positivity bias (higher learning rate after positive compared to negative outcomes) and the confirmation bias (higher learning rate after a choice was confirmed). While such effects can lead to biases in certain contexts, they might have evolved to strengthen the robustness of learning and decision-making. One line of research argues that positivity and confirmation biases lead to increased differences in reward estimates between options^{6,7,8}. For example, when the true underlying reward probability of a favorable option is 0.6, and the probability of the worse option is 0.4, such biases can increase the subjective difference between them so that, for example, the favorable option has a subjective probability of 0.7. Such an increased expected-value difference can lead to better discriminability between options. This would particularly optimize choice performance when choices are noisy or stochastic, that is, the decision-maker sometimes chooses a sub-optimal option due to random influences, where increased value differences between options would lead to more frequent choices in favor of the best option.

A related line of research shows that asymmetries in responses to positive and negative outcomes might alternatively be computed at the level of the decision-making policy instead of during learning via the learning rate^{9,10}. According to the opponent actor learning (OpAL) model, it might be normative to overemphasize preference differences between different actions¹⁰. One crucial advantage of this increased discrimination between actions is that choice behavior can be efficiently tuned to the reward environment. Particularly in lean reward environments in which it is challenging to find the best option based on exploration, this policy outperforms strategies without biased policy values. To conclude, both learning-rate and choice biases on the policy level might ultimately serve normative purposes, and as they are not mutually exclusive, might co-exist.*

- The computational psychiatry section is missing some relevant literature. For example, in the “autism spectrum disorder”, the way the paragraph is written it seems that only a few studies have been done on the matter. Instead, there is a huge literature on the topic (the authors may be aware of the literature, is just that a reader not familiar with it may get the wrong impression). For a very good review/meta-analysis on the topic, see Chrysaitis and Series, 2023.

We agree that there are more relevant studies on ASD, and in the last few years, there has been an interesting debate about the Bayesian brain in ASD. Therefore, we have substantially updated the ASD section. We now start with a brief overview about a theoretical debate about where ASD deficits might stem from in terms of Bayesian inference (altered priors vs. likelihoods). Subsequently, we discuss modeling literature that supports or argues against a Bayesian account of ASD (page 15-16):

ASD is a neurodevelopmental disorder with a diverse range of symptoms including difficulties in social interaction and communication, restricted and repetitive behavior, and increased attention to detail¹¹. A prominent hypothesis on the computational underpinnings of ASD symptoms holds

that deficits in perception, learning, and social cognition result from an imbalance of the precision ascribed to sensory evidence relative to prior beliefs^{12,13,14}. Accordingly, individuals with ASD would expect highly precise sensory input so that newly arriving information overrules prior beliefs during belief updating, akin to a higher learning rate. Currently, the evidence for this hypothesis is mixed, and we here provide a selective overview of modeling studies examining learning and decision-making in ASD (see also systematic and comprehensive review of studies across different cognitive domains by Chrysaitis and Seriès¹⁵).

Some studies are overall in line with the hypothesis of an imbalance in the impact of prior beliefs and new information on belief updating in ASD. One study shows that individuals reporting increased attention to detail show overlearning in environments with risk and change points¹⁶. From a normative perspective, this finding suggests that autistic individuals underestimate the magnitude of risk, thereby assuming that outcomes are less heavily distorted by random variability and treating prediction errors as more reliable teaching signals. This result aligns with the intuition that autistic individuals integrate fewer outcomes during learning, reflecting a strong focus on local stimulus details rather than the global average over the history of observations.

Moreover, in reversal learning, autistic individuals might overlearn about the frequency of environmental changes¹⁷. Based on the HGF, it has been suggested that their beliefs about environmental volatility fluctuate too strongly in response to new outcomes, thereby preventing accurate representations of how much the environment changes. A more recent study with a similar reversal-learning task is partly consistent with these results and found higher overall learning rates in children and younger adults with ASD compared to control subjects. These results were accompanied by higher perseveration rates in ASD participants¹⁸.

Other studies do not support the hypothesis of an imbalance between prior beliefs and newly arriving sensory information. Two studies with reversal-learning tasks failed to find evidence of learning-rate differences between ASD and control groups^{19,20}. Another study did not find significant group differences in volatility estimates between an ASD and control group but hinted at potential subgroup differences in ASD, which would need to be confirmed independently²¹. Moreover, a study in the general population did not find systematic evidence of a correlation between autistic traits and learning rates²². Finally, a developmental study comparing children with ASD and a matched control group suggests similar learning-rate adjustments to different degrees of volatility²³.

Taken together, evidence of systematic learning-rate differences between groups indicating an imbalance between the influences of prior beliefs and new information on belief updating is mixed. We note that some of the studies that do not support the imbalance hypothesis found performance differences between groups, but such differences seem to originate independently of learning rates^{20,22}. Crucially, the studies were obtained with different tasks, participant populations, and modeling frameworks. This highlights the need for future work to rigorously examine the psychometric properties of the measures used and to systematically compare tasks to ensure the robustness and generalizability of the results.

- Small suggested change: In the abstract, instead of “Learning allows humans and animals” change to “Learning allows humans and other animals”.

We have added "other animals", in line with the reviewer's suggestion.

Reviewer 2

This review article discusses how humans and animals learn under different forms of uncertainty, such as perceptual uncertainty, risk, and environmental change, using normative Bayesian models as a framework. The authors explore how learning biases arise from either misaligned priors or computational approximations, and provide insights into psychiatric disorders.

I believe this review is highly valuable and useful for those interested in computational perspectives on perception, learning and behaviour. It provides a well-organised summary of topics ranging from traditional frameworks to recent advances, making it a useful resource. Before publication, I would like to ask for a few clarifications, which I have outlined below.

Comment #1: This review primarily focuses on normative Bayesian models, which infer environmental states through Bayesian inference based on an environmental model. However, particularly in the discussions on the relationship between learning rates and psychiatric disorders, many cited studies appear to use parameter estimation from simpler reinforcement learning models (e.g., the Rescorla-Wagner model), which often assume the absence of perceptual uncertainty. It would be helpful to briefly mention these models and clearly inform readers that not all referenced studies were conducted using Bayesian models. Additionally, it would be beneficial to include an explanation on how behavioral data is fitted to parameters, as this may not be familiar to all readers.

It's correct that we cite several studies that relied on reinforcement learning. To clarify that reinforcement-learning models can approximate Bayesian computations and help us understand learning-rate differences between clinical and healthy groups, we have added the following sentences on page 3 (lines 7-10):

We incorporate studies that directly use Bayesian computational models examining the impact of uncertainty on learning. We also discuss work that relies on reinforcement-learning approaches^{24,25,26}, which can, under specific conditions, be used to approximate principles of Bayesian learning (Box 1.)

Moreover, to ensure that non-expert readers understand the main ideas behind reinforcement learning, the Rescorla-Wagner model, and the relationship to Bayesian models, we have added a box on the "Basics of reinforcement learning" (page 6):

Reinforcement learning is an overarching framework for solving learning and decision-making problems to maximize reward over time²⁴. In psychology and neuroscience, the most popular reinforcement-learning model is the Rescorla-Wagner (RW) model, which provides a simple and elegant approach to associative learning. The basic idea is that the model computes the difference between an obtained reward r_t on a given trial and the expected reward or expected value v_t , known as the prediction error δ_t :

$$\delta_t = r_t - v_t \tag{2}$$

and uses the prediction error to update the expected value v_{t+1} for the next trial

$$v_{t+1} = v_t + \alpha \cdot \delta_t \quad (3)$$

where α denotes the learning rate determining the impact of the prediction error on the expected-value update. The interpretation of the learning rate can be illustrated based on the rearranged update equation $v_{t+1} = (1 - \alpha) \cdot v_t + \alpha \cdot r_t$, where it becomes clear that α controls the influence of the most recent outcome r_t relative to the prior expected value v_t . Greater learning rates (~ 1) yield a stronger influence of the most recent reward, while lower values (~ 0) lead to a stronger influence of the previous expected value.

The RW model inspired the formulation of several widely used Bayesian models in psychology and neuroscience. For example, the reduced Bayesian model (RBM)²⁷ and Hierarchical Gaussian Filter (HGF)²⁸ use update equations similar to eq. (3), where the learning rate is usually a dynamic parameter that changes from trial to trial, in contrast to the RW model with constant α . This similarity allows researchers to approximate Bayesian learning with the RW model under certain circumstances. For example, based on the RW model, one can examine learning-rate differences between different environmental conditions, such as frequently changing and stable environments, where a Bayesian perspective predicts higher learning rates in environments with more frequent changes^{29,30,31}.

Finally, in line with the reviewer’s suggestion, we have added a box on the topic of parameter estimation (page 18):

Computational models can be interpreted as quantitative hypotheses about cognitive processes underlying task behavior. We can develop different models, where each represents a distinct hypothesis, and compare such models to each other in light of the observed data. To do so, researchers often rely on parameter estimation or "model fitting" to find the set of parameters that best describes the participants’ behavioral data, such as the learning rate or the consideration of uncertainty for learning.

We typically distinguish between a learning model like the Rescorla-Wagner model or the reduced Bayesian model (RBM) formalizing the learning process itself and the observation model linking the learning model’s predictions and the observed empirical data. The purpose of the observation model is to assign probabilities to the measured data conditional on the predictions of the learning model. To find the best-fitting model parameters, we can rely on a number of available methods. The most common method is maximum-likelihood estimation. The combination of the learning and observation model defines a likelihood function, and maximum-likelihood estimation allows us to compute the most probable parameter values, that is, the values that maximize the likelihood function. For example, when estimating free parameters of the RBM, the likelihood function might be defined by

$$p^\varepsilon(a_{1:T}|x_{1:T}) = N(a_{1:T}; \hat{a}_{1:T}, \varepsilon^2) \quad (4)$$

where a denotes the empirical data (in this case, the belief updates of the participant) in response to the presented outcomes x , and \hat{a} denotes the model’s belief updates. ε is a motor-noise parameter

that accounts for the residuals arising from deviations between participant and model behavior, and T refers to the total number of trials. In this example, the RBM corresponds to the learning model based on which we obtain the model’s belief updates \hat{a} , and the Gaussian in eq. (4) is the observation model linking the RBM to the empirical data (here a). The motor-noise parameter ε accounting for residuals due to imprecise motor control has to be estimated (panel a shows three values of ε for an example trial t). Panel b shows the log-likelihood of a range of parameter values (blue curve), and maximum-likelihood estimation allows us to find the best parameter value with maximum likelihood (black dot). Beyond the parameters of the observation model, we could use the method of maximum likelihood to estimate parameters of the RBM (i.e., the learning model), such as the influence of change-point probability on learning or the underestimation of uncertainty^{32,33}.

Finally, comprehensive modeling requires additional analyses and methods, including the systematic comparison of alternative models and different model-validation analyses like parameter and model recovery. Moreover, there are alternative approaches to parameter estimation available such as hierarchical Bayesian parameter estimation. Excellent guidelines and tutorials are provided by references^{34,35}.

Comment #2: Regarding The Origin of Learning Bias (p.15-): I understand that when the number of samples in MCMC is finite, estimation biases can occur. However, if the sampling process is properly conducted, these biases would not consistently skew in one direction but instead fluctuate randomly. This raises the question of how specific directional biases observed in psychiatric disorders can be explained by this mechanism. Perhaps I am overlooking something, but if this issue has not yet been addressed, I would appreciate a discussion on this point.

We are sorry that we failed to communicate this point more clearly. We have added the following explanation on why directional biases can occur within sampling models (page 21, lines 11-17):

The anchoring bias emerges in particular when the model only draws a limited amount of samples and concludes the sampling process before the new samples can fully compensate for the influence of the external anchor. In typical applications of MCMC, biases due to the start point of the sampling process are usually eliminated by discarding the initial samples, known as the "burn-in period". However, sampling models with anchoring bias, including our own model, explicitly assume that burn-in samples are not discarded, yielding the anchoring bias^{33,36}.

Minor points:

We are going to address the two following comments together:

p.7, Line 29- “Human participants seem to regulate their learning rates accordingly and tend to use higher learning rates when estimation uncertainty is high. . .” I expected that lower learning rates would be used in such cases, rather than higher ones. If my understanding is incorrect, I would appreciate a brief clarification to make this point clearer.

p.11 Box 1 It might be helpful to include a brief and intuitive explanation of why equations (1) and (2) take their specific forms. This would aid readers in understanding the underlying rationale behind these formulations.

We have substantially updated the box on the reduced Bayesian model (RBM). Now we provide more detailed and intuitive explanations for the model equations and clarify why higher estimation uncertainty leads to higher learning rates (page 18):

The reduced Bayesian model (RBM) provides a way to study how humans adaptively update their beliefs under uncertainty and in changing environments^{27,37}. The model simplifies a full Bayesian solution to change-point detection³⁸ for reduced computational complexity and biological plausibility. The update equation is inspired by the Rescorla-Wagner (RW) model:

$$\mu_{t+1} = \mu_t + \alpha_t \cdot \delta_t \quad (5)$$

where μ_t denotes the model’s belief (e.g., about the taste of a pastry), and the prediction error is defined by:

$$\delta_t = x_t - \mu_t \quad (6)$$

expressing the difference between outcome x_t and belief μ_t . The RBM extends beyond the RW model because it relies on a dynamic learning rate that can change from trial to trial depending on uncertainty and environmental changes:

$$\alpha_t = \omega_t + \tau_t - \tau_t \cdot \omega_t \quad (7)$$

The learning rate is composed of two key terms. Change-point probability ω_t indicates the probability that the last outcome x_t was generated by a change point ($c_t = 1$) under consideration of the prior probability of a change point (hazard rate, h):

$$\omega_t = \frac{p(x_t|c_t = 1) \cdot h}{p(x_t|c_t = 0) \cdot (1 - h) + p(x_t|c_t = 1) \cdot h} \quad (8)$$

The parameter is computed based on Bayes’ rule, and the learning rate increases with higher change-point probability. This ensures that the model rapidly adjusts its beliefs after a change in the environment. Finally, relative uncertainty τ_t reflects a combination of risk σ^2 and estimation uncertainty σ_t^2 :

$$\tau_t = \frac{\sigma_t^2}{\sigma_t^2 + \sigma^2} \quad (9)$$

Relative uncertainty increases in response to a peak in change-point probability and slowly decays as a function of time. The main idea is that the learner’s uncertainty over the current belief (estimation uncertainty σ_t) decreases relative to the total uncertainty (sum of estimation uncertainty and risk σ^2). Therefore, a decrease in estimation uncertainty over trials reflects that the model’s belief is getting more and more certain. This leads to a reduction in the learning rate, yielding a weaker influence of new outcomes of the belief update.

p.15 Line 26 “the denominator requires solving integrals that are infeasible or costly based on numerical methods.” - I am unclear about why calculating this denominator is necessary. If the goal is to determine the relative plausibility of different values of μ , it seems the denominator may not be required, as normalization into probabilities isn’t necessary (e.g., as in MAP estimation). It would be helpful if this point could be clarified.

The reviewer’s intuition that computing the denominator is not necessary when approximate methods are used is correct. The starting point of our argument was that exact Bayesian inference based on analytic update equations requires the denominator of Bayes’ rule for normalization. This often involves integrals that are analytically infeasible, and numerical approximations to these integrals might be computationally costly. As pointed out by the reviewer, several approximation schemes don’t require the denominator, which reduces computational costs. Indeed, MCMC – the approximation method that we proposed as a biologically plausible approach – does not require computing the denominator. In our updated paragraph, we clarify that we meant to say that solving the denominator is required for analytical solutions or when using available numerical approximations for these analytical solutions. Moreover, we have clarified that biologically plausible approximation schemes should not require explicitly solving the denominator (page 19, lines 20-26):

When doing Bayesian inference, we are interested in the posterior distribution $p(\mu|x)$. Computing the posterior based on analytic update equations can be challenging when the denominator requires solving integrals that are infeasible or when computations are costly based on numerical approximation methods. Therefore, if the nervous system has evolved to perform normative computations, it has to rely on smart approximations at lower computational costs that do not require explicitly computing the denominator with intractable integrals.

Figure 1a (p.4) This might stem from cultural differences and may not need much attention, but as a Japanese reader, I found the depiction of the rice bowls a bit strange. The image seems to resemble plain bowls of rice without toppings, which is not typically something one would choose between at a restaurant. Additionally, in Japanese culture, standing chopsticks upright in a bowl is a gesture reserved for offerings to the deceased and is considered highly inauspicious and inappropriate in other contexts.

This is very interesting and good to know. We have updated the example and now use different pastries instead of rice bowls.

That said, the authors' narrative is very clear, and the fundamental storyline is well-constructed. I believe it can remain as it is without any major changes.

Thank you for your kind and encouraging feedback.

References

- [1] Courville, A. C., Daw, N. D., & Touretzky, D. S. (2006). Bayesian theories of conditioning in a changing world. *Trends in Cognitive Sciences*, *10*(7), 294–300. <https://doi.org/10.1016/j.tics.2006.05.004>
- [2] FeldmanHall, O., & Nassar, M. R. (2021). The computational challenge of social learning. *Trends in Cognitive Sciences*, *25*(12), 1045–1057. <https://doi.org/10.1016/j.tics.2021.09.002>
- [3] FeldmanHall, O., & Shenhav, A. (2019). Resolving uncertainty in a social world. *Nature Human Behaviour*, *3*(5), 426–435. <https://doi.org/10.1038/s41562-019-0590-x>
- [4] Bossaerts, P. (2009). What decision neuroscience teaches us about financial decision making [Publisher: Annual Reviews]. *Annual Review of Financial Economics*, *1*, 383–404. <https://doi.org/10.1146/annurev.financial.102708.141514>
- [5] Palminteri, S., & Lebreton, M. (2022). The computational roots of positivity and confirmation biases in reinforcement learning. *Trends in Cognitive Sciences*, *26*(7), 607–621. <https://doi.org/10.1016/j.tics.2022.04.005>
- [6] Cazé, R. D., & van der Meer, M. A. A. (2013). Adaptive properties of differential learning rates for positive and negative outcomes. *Biological Cybernetics*, *107*(6), 711–719. <https://doi.org/10.1007/s00422-013-0571-5>
- [7] Lefebvre, G., Lebreton, M., Meyniel, F., Bourgeois-Gironde, S., & Palminteri, S. (2017). Behavioural and neural characterization of optimistic reinforcement learning. *Nature Human Behaviour*, *1*(4), 0067. <https://doi.org/10.1038/s41562-017-0067>
- [8] Lefebvre, G., Summerfield, C., & Bogacz, R. (2022). A normative account of confirmation bias during reinforcement learning. *Neural Computation*, *34*(2), 307–337. https://doi.org/10.1162/neco_a_01455
- [9] Collins, A. G. E., & Frank, M. J. (2014). Opponent actor learning (OpAL): Modeling interactive effects of striatal dopamine on reinforcement learning and choice incentive. *Psychological Review*, *121*(3), 337–366. <https://doi.org/10.1037/a0037015>
- [10] Jaskir, A., & Frank, M. J. (2023). On the normative advantages of dopamine and striatal opponency for learning and choice (M. Liljeholm, J. I. Gold, & M. Liljeholm, Eds.) [Publisher: eLife Sciences Publications, Ltd]. *eLife*, *12*, e85107. <https://doi.org/10.7554/eLife.85107>
- [11] American Psychiatric Association. (2013). *Diagnostic and statistical manual of mental disorders: DSM-5* (5th ed.). <https://doi.org/10.1176/appi.books.9780890425596>
- [12] Lawson, R. P., Rees, G., & Friston, K. J. (2014). An aberrant precision account of autism [Publisher: Frontiers]. *Frontiers in Human Neuroscience*, *8*. <https://doi.org/10.3389/fnhum.2014.00302>
- [13] Pellicano, E., & Burr, D. (2012). When the world becomes ‘too real’: A bayesian explanation of autistic perception [Publisher: Elsevier]. *Trends in Cognitive Sciences*, *16*(10), 504–510. <https://doi.org/10.1016/j.tics.2012.08.009>
- [14] Brock, J. (2012). Alternative bayesian accounts of autistic perception: Comment on pellicano and burr [Publisher: Elsevier]. *Trends in Cognitive Sciences*, *16*(12), 573–574. <https://doi.org/10.1016/j.tics.2012.10.005>
- [15] Chrysaitis, N. A., & Seriès, P. (2023). 10 years of bayesian theories of autism: A comprehensive review. *Neuroscience & Biobehavioral Reviews*, *145*, 105022. <https://doi.org/10.1016/j.neubiorev.2022.105022>

- [16] Nassar, M. R., & Troiani, V. (2021). The stability flexibility tradeoff and the dark side of detail. *Cognitive, Affective, & Behavioral Neuroscience*, *21*(3), 607–623. <https://doi.org/10.3758/s13415-020-00848-8>
- [17] Lawson, R. P., Mathys, C. D., & Rees, G. (2017). Adults with autism overestimate the volatility of the sensory environment. *Nature Neuroscience*, *20*(9), 1293–1299. <https://doi.org/10.1038/nn.4615>
- [18] Crawley, D., Zhang, L., Jones, E. J. H., Ahmad, J., Oakley, B., San José Cáceres, A., Charman, T., Buitelaar, J. K., Murphy, D. G. M., Chatham, C., Den Ouden, H., Loth, E., & the EU-AIMS LEAP group. (2020). Modeling flexible behavior in childhood to adulthood shows age-dependent learning mechanisms and less optimal learning in autism in each age group. *PLoS Biology*, *18*(10), e3000908. <https://doi.org/10.1371/journal.pbio.3000908>
- [19] Sapey-Triomphe, L.-A., Temmerman, J., Puts, N. A. J., & Wagemans, J. (2021). Prediction learning in adults with autism and its molecular correlates. *Molecular Autism*, *12*(1), 64. <https://doi.org/10.1186/s13229-021-00470-6>
- [20] Sapey-Triomphe, L.-A., Weilhhammer, V. A., & Wagemans, J. (2022). Associative learning under uncertainty in adults with autism: Intact learning of the cue-outcome contingency, but slower updating of priors. *Autism*, *26*(5), 1216–1228. <https://doi.org/10.1177/13623613211045026>
- [21] Kreis, I., Biegler, R., Tjelmeland, H., Mittner, M., Reitan, S. K., & Pfuhl, G. (2021). Overestimation of volatility in schizophrenia and autism? a comparative study using a probabilistic reasoning task [Publisher: Public Library of Science]. *PLOS ONE*, *16*(1), e0244975. <https://doi.org/10.1371/journal.pone.0244975>
- [22] Goris, J., Silvetti, M., Verguts, T., Wiersema, J. R., Brass, M., & Braem, S. (2021). Autistic traits are related to worse performance in a volatile reward learning task despite adaptive learning rates [Publisher: SAGE Publications Ltd]. *Autism*, *25*(2), 440–451. <https://doi.org/10.1177/1362361320962237>
- [23] Manning, C., Kilner, J., Neil, L., Karaminis, T., & Pellicano, E. (2017). Children on the autism spectrum update their behaviour in response to a volatile environment. *Developmental Science*, *20*(5), e12435. <https://doi.org/10.1111/desc.12435>
- [24] Sutton, R. S., & Barto, A. G. (1998). *Reinforcement learning: An introduction*. MIT Press.
- [25] Daw, N. D. (2014). Advanced reinforcement learning. In *Neuroeconomics* (pp. 299–320). Elsevier. <https://doi.org/10.1016/B978-0-12-416008-8.00016-4>
- [26] Dayan, P., & Daw, N. D. (2008). Decision theory, reinforcement learning, and the brain. *Cognitive, Affective, & Behavioral Neuroscience*, *8*(4), 429–453. <https://doi.org/10.3758/CABN.8.4.429>
- [27] Nassar, M. R., Wilson, R. C., Heasley, B., & Gold, J. I. (2010). An approximately bayesian delta-rule model explains the dynamics of belief updating in a changing environment. *Journal of Neuroscience*, *30*(37), 12366–12378. <https://doi.org/10.1523/JNEUROSCI.0822-10.2010>
- [28] Mathys, C. D. (2011). A bayesian foundation for individual learning under uncertainty. *Frontiers in Human Neuroscience*, *5*. <https://doi.org/10.3389/fnhum.2011.00039>
- [29] Behrens, T. E. J., Woolrich, M. W., Walton, M. E., & Rushworth, M. F. S. (2007). Learning the value of information in an uncertain world. *Nature Neuroscience*, *10*(9), 1214–1221. <https://doi.org/10.1038/nn1954>

- [30] Browning, M., Behrens, T. E., Jocham, G., O'Reilly, J. X., & Bishop, S. J. (2015). Anxious individuals have difficulty learning the causal statistics of aversive environments. *Nature Neuroscience*, *18*(4), 590–596. <https://doi.org/10.1038/nn.3961>
- [31] Gagne, C., Zika, O., Dayan, P., & Bishop, S. J. (2020). Impaired adaptation of learning to contingency volatility in internalizing psychopathology. *eLife*, *9*, e61387. <https://doi.org/10.7554/eLife.61387>
- [32] Nassar, M. R., Bruckner, R., Gold, J. I., Li, S.-C., Heekeren, H. R., & Eppinger, B. (2016). Age differences in learning emerge from an insufficient representation of uncertainty in older adults. *Nature Communications*, *7*(1), 11609. <https://doi.org/10.1038/ncomms11609>
- [33] Bruckner, R., Nassar, M. R., Li, S.-C., & Eppinger, B. (2020). Differences in learning across the lifespan emerge via resource-rational computations [Accepted for publication in Psychological Review]. *Psychological Review (in Press)*. <https://doi.org/10.31234/osf.io/nh9bq>
- [34] Daw, N. D. (2011). Trial-by-trial data analysis using computational models. In *Decision making, affect, and learning: Attention and performance XXIII* (pp. 3–38).
- [35] Wilson, R. C., & Collins, A. G. E. (2019). Ten simple rules for the computational modeling of behavioral data. *eLife*, *8*, e49547. <https://doi.org/10.7554/eLife.49547>
- [36] Lieder, F., Griffiths, T. L., Huys, Q. J. M., & Goodman, N. D. (2018). The anchoring bias reflects rational use of cognitive resources. *Psychonomic Bulletin & Review*, *25*(1), 322–349. <https://doi.org/10.3758/s13423-017-1286-8>
- [37] Wilson, R. C., Nassar, M. R., & Gold, J. I. (2010). Bayesian online learning of the hazard rate in change-point problems. *Neural Computation*, *22*(9), 2452–2476. https://doi.org/10.1162/NECO_a_00007
- [38] Adams, R. P., & MacKay, D. J. C. (2007, October 19). Bayesian online changepoint detection. <http://arxiv.org/abs/0710.3742>